# Cross-cutting scenarios and strategies for designing decarbonization pathways in the transport sector toward carbon neutrality

Runsen Zhang [1,2] & Tatsuya Hanaoka[2]

The transport sector will play a pivotal role in achieving China's carbon neutrality goal by 2060. This study develops a regional transport-energy integrated model to analyze the long-term pathways and strategies toward the carbon-neutral ground transport sector in China. A set of scenarios are created to identify the effectiveness and feasibility of low-carbon policy measures based on the well-known transport strategies within the Avoid–Shift–Improve framework. Our simulations shed light on synergistic coupling and trade-offs among different strategies and instruments for prescribing a desirable mix of policy measures that maximize the synergies and minimize the trade-offs. Here, we show that a region-specific policy package designed from a balanced perspective under the Avoid–Shift–Improve framework has the potential to realize a deep decarbonization in the transport sector and will greatly assist in achieving China's carbon neutrality by 2060.

[1] Graduate School of Advanced Science and Engineering, Hiroshima University, Higashihiroshima, Japan. [2] Social Systems Division, National Institute for Environmental Studies, Tsukuba, Japan. ✉email: rzhang@hiroshima-u.ac.jp

Driven by fossil-fuel-dominated industrialization and rapid urbanization, China is burdened with a tremendous pressure to reduce carbon emissions. As the largest emitter of carbon dioxide ($CO_2$) in the world, China accounted for approximately 30% of global total carbon emissions in 2018, representing twice the emissions produced by the world's second largest emitter, the United States, accounting for approximately 15% of the global total[1]. Whether and how China can achieve its emission reduction goal will have significant impacts and implications on the reduction of global $CO_2$ emissions, and China's willingness and ability to transition toward a low-carbon economy will play an important role in the mitigation of global climate change[2]. In response to the growing need to explore low-carbon options for economic development, China declared a carbon neutral goal to be achieved before 2060 at the General Debate of the 75th Session of the United Nations General Assembly in September 2020[3]. The government of China has also announced a goal of carbon peaking by 2030 to pave the way for achieving its long-term commitment to carbon neutrality by 2060. Due to China's ambitious short- and long-term goals, the so called 30/60 targets, the concept of carbon neutrality has become one of the underpinning pillars of the nationwide transition to a low-carbon economy across all sectors by enhancing emission reduction efforts.

Driven by economic development and unprecedented urbanization, transportation has become one of China's fastest growing economic sectors, with the turnover volumes of passenger and freight transport increasing rapidly since 2000. Passenger turnover rose from 1.23 trillion person-km in 2000 to 3.53 trillion person-km in 2019, while freight turnover increased from 4.43 to 19.94 trillion ton-km during the same period[4]. As a primary sector providing a fundamental service and boosting economic development, the rapidly growing demand for transport activities has generated increased pressure on the environment[5,6]. The transport sector is a significant contributor of $CO_2$ emissions in China due to the combustion of fossil fuels and will therefore play an important role in mitigating climate change[7]. China's transport sector contributed 901 million tons to $CO_2$ emissions in 2019, accounting for 10% of all $CO_2$ emissions nationwide. From 1990 to 2019, there has been a nearly ten-fold increase in transport-related $CO_2$ emissions from 94 million tons in 1990 alongside the vigorous development of the Chinese transport industry[1]. Considering the growing incomes, accelerating urbanization, and increasing car ownership in China, passenger and freight turnover will continue to proliferate over the coming decades, which will result in increased energy consumption and corresponding negative environmental impacts, e.g., increased $CO_2$ emissions[8]. Because passenger and freight transport activities are a major consumer of fossil fuels and a substantial source of carbon emissions, the transport sector is becoming a key sector for China to focus on if the carbon neutrality target is to be met before 2060.

The role of the transport sector in climate change mitigation and the corresponding low-carbon policies have attracted the attention of both climate scientists and transport planners globally. Integrated assessment models (IAMs) that incorporate transport factors such as mode choice and individual vehicle technologies have been used to project the future service demand, energy use, and emissions from transport[9–14]. Some global scenario studies have already discussed the contribution of the transport sector to climate change mitigation, particularly the long-term pathways to reach the deep decarbonization of transportation produced by IAMs[15–17]. Previous studies have examined the impacts of low-carbon policy measures, such as the adoption of more efficient vehicles and alternative sources of energy for vehicles (e.g., electricity, biofuel, and hydrogen)[18–20]. In transport planning research, transport models are widely applied to devise low-carbon transport management and planning strategies. An approach, known as Avoid–Shift–Improve (A-S-I), serves to structure policy measures to reduce the environmental impact of transport and thereby promote low-carbon transport development[21–24]. "Avoid" refers to the need to improve the efficiency of the transport system as a whole, through the transport-oriented and compact development of cities. "Shift" signifies a modal shift from the most energy consuming and polluting transport mode toward more environmentally friendly modes. "Improve" focuses on vehicle and fuel efficiency as well as on optimization of the operational efficiency of transport.

In global scenario studies, IAM modelers have highlighted the potential of the "Improve" strategy by reducing the energy consumption needed for a unit of transport demand and decreasing the carbon intensity of fuels consumed by the transport sector. In contrast, urban planners and transport experts are more concerned about how the "Avoid" and "Shift" strategies such as land use regulations, compact city, mass-transit development, and working from home will help design a low-carbon transport system. Methodologically, on the one hand, global IAMs depict an overall picture of energy consumption that appears plausible to energy policymakers and climate change scientists, but land use planning, infrastructure policies, and behavioral factors are hardly modeled in the current-generation global IAMs. On the other hand, transport models with sophisticated behavioral descriptions and a high spatial resolution can provide a significantly more concrete answer to urban transport problems, but they often simplify the representations of the energy system and lack a long-term integrated assessment of cross-sectoral effects. Therefore, to identify the long-term pathways toward the carbon-neutral transport sector in China, we need to develop an integrated framework embracing all possible options according to the A-S-I approach, which in general has not been modeled explicitly in the current-generation IAMs.

In this work, to capture the regional characteristics of transport energy use rather than an aggregated overview at the national level in China, we developed a transport energy model by integrating a transport model and an energy system model to project the future energy consumption and emissions of China's ground transport sector in 31 provinces. The transport model was established to offer spatial flexible and temporally dynamic simulations of transport demand and modal choices. It was coupled with a bottom-up optimization energy system model, with a detailed technology selection framework to detect the optimal technology and energy mix, and determine the corresponding emissions in the transport sector. The transport model passed the mode-specific service demand to the energy system model for estimating the future technology mix transition, energy consumption, and $CO_2$ emissions, while the technology mix and costs were fed back to the transport model to re-calculate the generalized transport cost with an updated technology mix. An iterative procedure was used to achieve the convergence of coupled models (see the Methods). The study focused on a 45-year period at 1-year intervals from 2015 up to 2060, the year of China's carbon neutral target.

## Results

**Cross-cutting scenarios based on the A-S-I framework**. Based on the A-S-I approach, we set up 12 scenarios for possible low carbon policy interventions to represent three strategies, avoid, shift, and improve, using four instruments: technology, regulation, information, and price (Fig. 1 and Methods). A business-as-usual (BaU) scenario, which presumed continuation of technological improvement at the current pace and maintenance of the existing transport and energy policies, was also designed to explore the default pathway toward the decarbonization of the

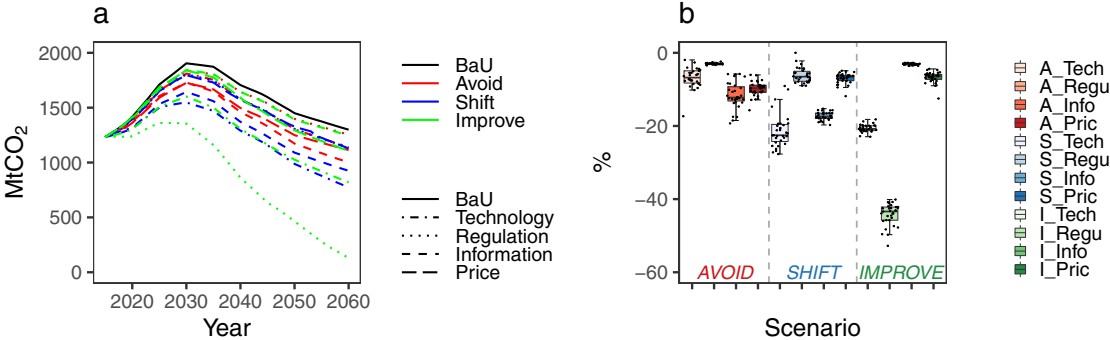

**Fig. 1 A-S-I matrix and policy examples.** Each strategy in the A-S-I matrix includes technological, regulatory, informatic, and price instruments.

**Fig. 2 CO₂ from the ground transport sector in China. a** Emission trajectories in China from 2015 to 2060. **b** Cumulative emission reduction potentials from 2015 to 2060 in 12 scenarios compared with the BaU scenario. In the box plot, data are presented as median and minimum/maximum values in 31 provinces. The box shows first and third quartile, and the whiskers are the maximum and minimum values.

transport sector and to measure the relative effects of different strategies and instruments on energy transition and emission reduction.

Figure 2a shows the projected $CO_2$ emissions produced by the ground transport sector in China from 2015 to 2060 under various A-S-I scenarios. Without implementing any policies in the BaU scenario, the $CO_2$ emissions peaked at 1904 Mt in 2030, followed by a steady decrease. Despite that the BaU scenario did not take into account additional policy measures, $CO_2$ emissions still exhibited a declining trend from 2030 to 2060. This was because energy-efficient technologies will gradually penetrate into the transport sector over the coming decades, and China's population is also expected to decrease after the mid-21st century. All 12 policy interventions designed based on the A-S-I matrix potentially caused substantial emission reductions from 2015 to 2060. The peak values of the $CO_2$ emission trajectories were lowered significantly, ranging from 1844 Mt (I_Info scenario) to 1361 Mt (I_Regu scenario), compared with the peak value under the BaU scenario of 1904 Mt in 2030. The I_Regu scenario presented the most substantial emission reduction in 2060, with 90% (1170 Mt) of the 1299 Mt in the BaU scenario reduced due to the regulatory instrument in the "Improve" strategy. The A_Regu scenario resulted in the lowest emission reduction (45 Mt) in 2060.

The scenario simulations showed that policy interventions would lead to different reduction potentials in the cumulative emissions during 2015–2060 across different strategies and instruments (Fig. 2b). The I_Regu scenario had the most substantial policy effectiveness, with 44% of cumulative $CO_2$ emissions being reduced from 2015 to 2060. The lowest emission reduction was attributed to the A_Regu scenario, in which approximately 3% of the cumulative $CO_2$ emissions were reduced. The policies created in the "Avoid" strategy generated the most

moderate reductions in cumulative emissions, with a range of 3% (A_Regu) to 12% (A_Info) from 2015 to 2060. The "Improve" strategy had the most dispersed policy effectiveness, with emission reductions of 3% and 44% in the I_Info and I_Regu scenarios, respectively.

**Pathways of the transport sector toward carbon neutrality.** The future pathways toward carbon-neutral transport were further analyzed by combining the 12 scenarios into three strategies and four instruments. All instrument scenarios under the same strategy category were combined as "Avoid", "Shift", and "Improve" scenarios, while the same instruments under different strategies were combined as "Technology", "Regulation", "Information", and "Price" scenarios. As shown in Fig. 3a, $CO_2$ emissions in 2060 decreased from 1299 Mt in the BaU scenario to 862, 466, and 72 Mt under the combined scenarios of "Avoid", "Shift", and "Improve", respectively. Meanwhile, scenarios combined according to the dimension of instruments showed that $CO_2$ emissions in 2060 would be reduced to 429, 44, 675, and 756 Mt due to the implementation of technological, regulatory, informatic, and price instruments, respectively. A decarbonization pathway toward a carbon-neutral transport sector was structured under the assumption that all policy interventions within the A-S-I approach would be carried out to achieve the maximum emission reduction potential toward carbon neutrality, with 81% (59 Gt) of $CO_2$ emissions being reduced from 2015 to 2060, with all behavioral changes and technological transformations taken into account. Among the 59 Gt reductions in cumulative $CO_2$ emissions during 2015 to 2060 required by the carbon neutral goal, "Avoid", "Shift", and "Improvement" contributed to 28%, 48%, and 69% of emission reductions, respectively, while 50%, 61%, 36%, and 40% of emission reductions were attributed to the

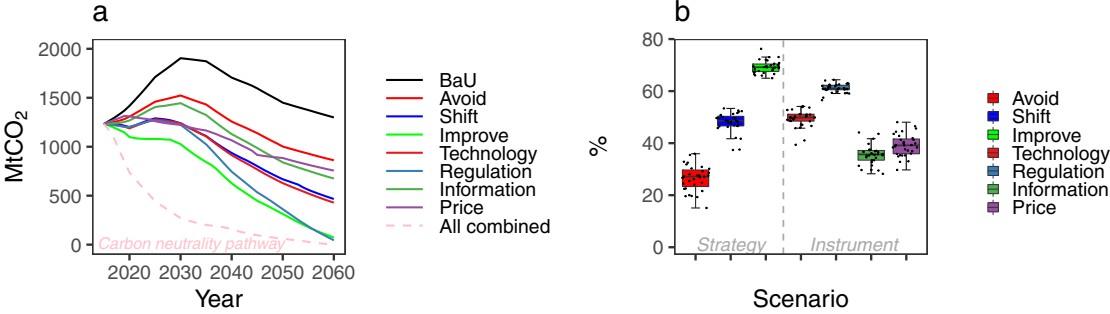

**Fig. 3 CO₂ emission pathways under different strategies and instruments. a** Emission trajectories under combined scenarios. **b** Contribution of three strategies and four instruments to achieving carbon neutrality by 2060. In the box plot, data are presented as median and minimum/maximum values in 31 provinces. The box shows first and third quartile, and the whiskers are the maximum and minimum values.

technological, regulatory, informatic, and price instruments, respectively (Fig. 3b).

We considered the key drivers of changes in CO₂ emissions, i.e., transport demand, energy intensity, and carbon intensity, to investigate how the different strategies and instruments would affect the emission reductions (Fig. 4). Without any policy measures in the BaU scenario, although the transport demand increased substantially from 2015 to 2060, the volume of the cubes that represented CO₂ emissions in 2060 was lower than the 2015 level for both passenger and freight transport. This was due to the significantly lower energy intensity and carbon intensity. Figure 4a, b shows that the "Avoid" strategy reduced the emissions by decreasing the transport demand, but there was no obvious change in the energy intensity or carbon intensity. In contrast, although the "Shift" and "Improve" strategies induced increases in transport demand, emissions decreased sharply due to the much lower energy intensity and carbon intensity. The carbon intensities decreased substantially in the "Improve" scenario due to the improvements in vehicle and energy technologies. As shown in Fig. 4c, while the informatic instrument could substantially reduce the passenger transport demand, energy intensity and carbon intensity only changed slightly compared with the BaU level. The regulatory instrument reduced the volume of cubes by lowering the carbon intensity most significantly. Freight transport demand was only slightly diminished by the price instrument, and the strongest emission reduction was attributed to the regulatory instrument, which greatly decreased the carbon intensity (Fig. 4d).

**Regional disparities in reduction effects by policy measures.** The impacts of different strategies and instruments on emission reductions varied spatially across China's 31 regions (Fig. 5). By implementing the "Avoid" strategy and the instruments of "Information" and "Price", western regions made a relatively weak contribution to emission reductions, while more substantial reduction potentials were apparent in eastern China, particularly the coastal metropolitan areas (e.g., Beijing, Shanghai, Jiangsu, Zhejiang, and Shandong). In contrast, the "Shift" strategy and instruments of "Technology" and "Regulation" presented the opposite pattern of policy effectiveness; namely, the southern and southwestern regions had high emission reduction contributions, whereas the eastern and northern regions had lower reduction potentials produced by the modal shift, technological developments, and regulations. The distribution of reduction potentials due to the "Improve" strategy displayed a center-periphery pattern, in which policy impacts in central regions were less than those in the peripheral regions, particularly the southwestern and coastal provinces. The correlations between GDP per capita and reduction potentials had the same patterns. Positive correlations between GDP per capita and reduction potentials were observed in the "Avoid", "Information", and "Price" scenarios, whereas the "Shift", "Technology", and

"Regulation" scenarios produced negative correlations. In the "Improve" scenario, no significant correlation was found between GDP per capita and reduction potentials, implying that the policy impacts of vehicle and fuel efficiency improvements were unrelated to local economic prosperity. Regional disparities in the impacts of each policy intervention on emission reductions indicated that policies on traffic management, information technology, and pricing instrument may be more suitable for implementation in developed regions, while policies for shifting modal structure, promoting new technologies, and releasing regulations offer promising options for developing regions.

**Economic costs of carbon-neutral transport.** While the policy interventions adopted based on the A-S-I approach from three strategies and four instruments effectively supported the achievement of the carbon neutral goal, the investment costs due to different strategies and instruments differed. Figure 6a shows that the cumulative investment costs in the BaU scenario would be reduced by implementing A-S-I strategies and instruments. The most substantial reduction was achieved by adoption of the "Avoid" strategy (97 trillion RMB) and the "Information" instrument (95 trillion RMB), while the "Improve" strategy and "Regulation" instrument had very limited effects on cost reduction, with a decrease of only 6 and 3 trillion RMB of investment costs, respectively. Moreover, the correlations of the contribution with carbon neutrality and investment cost changes revealed that a higher emission reduction potential generally resulted in higher investment costs (Fig. 6b). The "Improve" strategy was the most effective way to stimulate an energy transition away from fossil fuels and to reduce CO₂ emissions from transport, but it also required more investment costs for advanced technology such as electric vehicles (EVs), fuel cell vehicles (FCVs), and intelligent transport systems. Although policies designed according to the "Avoid" strategy would have only a moderate influence on the energy mix and emission profiles, they would not create additional cost pressures, with potential consequences for economic feasibility and competitiveness.

**Benefits of comprehensive policy packages.** Our findings neither depreciated the positive effects of transport demand management nor unilaterally overstated the contribution of technological improvements to the decarbonization of the transport sector. We rather highlighted a balanced perspective to detect the trade-offs among strategies and instruments as they impact upon the extent to which the ultimate intended goals or outcomes. Although the "Improve" strategy provides a plausible solution due to its most substantial positive effects on emission reductions, the improvement of fuel economy due to the "Improve" strategy was inclined to generate more travel demand, which may negatively affect the

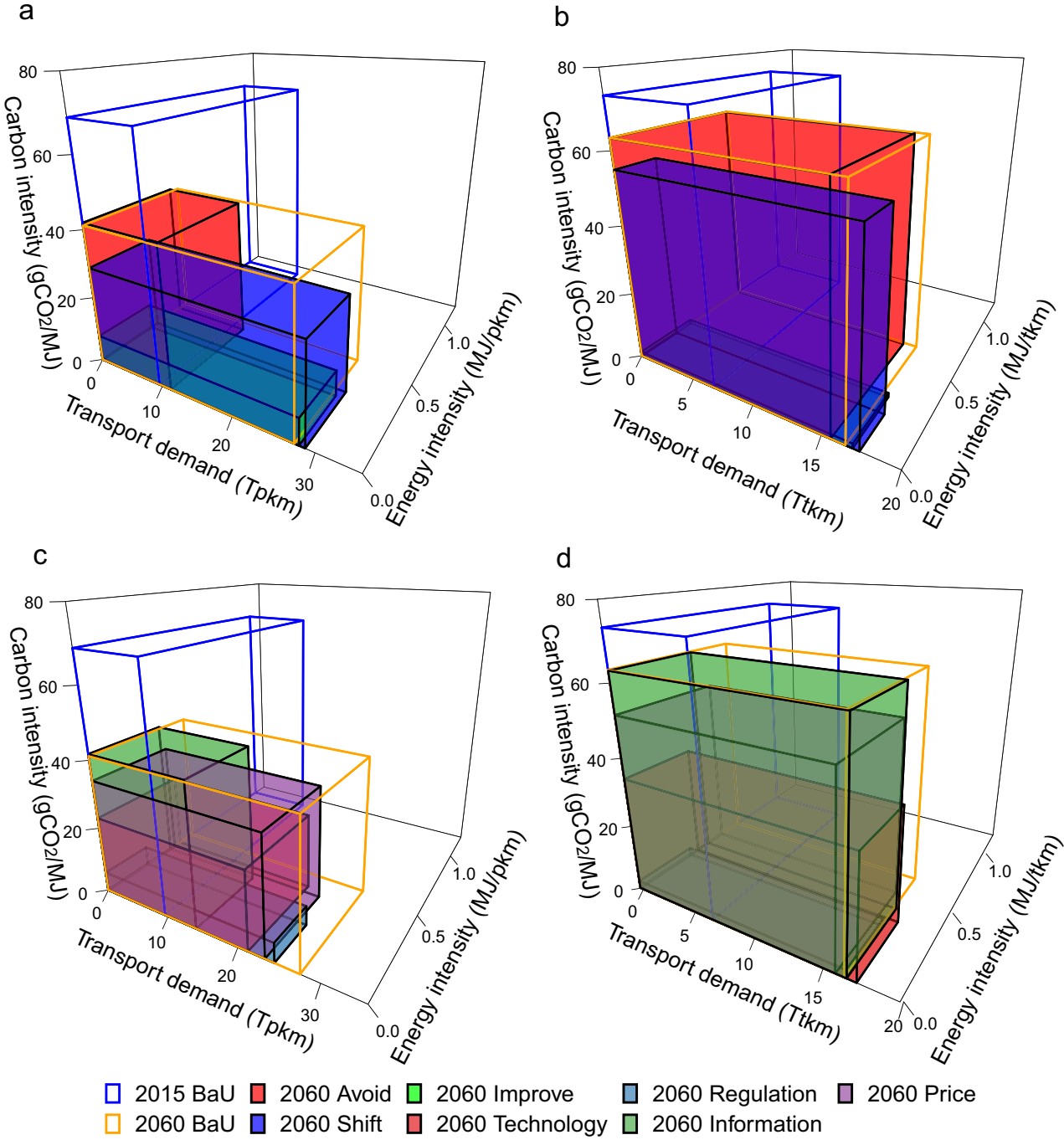

**Fig. 4 CO₂ emissions, transport demand, energy intensity, and carbon intensity in 2015 and 2060 under different strategies and instruments.**
**a** Passenger transport under BaU and three strategies. **b** Freight transport under BaU and three strategies. **c** Passenger transport under BaU and four instruments. **d** Freight transport under BaU and four instruments.

mitigation effects, particularly the demand reduction effects contributed by the "Avoid" strategy. Furthermore, policy effectiveness of the "Shift" strategy that encourages travelers to shift from private motorized vehicles to public transport may be weakened to some extent by the "Improve" strategy. For example, the petrol car ban and EV subsidies would increase the private car travel mode choice, which may negatively influence the use of public transport promoted by the policies prioritizing high-speed rail and buses.

While the trade-offs describe potential conflicts among different A-S-I strategies and instruments, policymakers must consider the synergistic coupling of various policy measures. The

"Avoid" strategy, e.g., land use planning and compact city form, can reduce travel demand by cutting down on long-distance trips but might also generate negative externalities (e.g., traffic congestion in the city center), which would offset the benefits produced by the "Avoid" strategy. Because the externalities of congestion could probably be alleviated through promotion of the public transport, the "Avoid" strategy needs to be implemented with the adoption of policy instruments designed according to the "Shift" strategy. In addition, compared to the behavioral interventions of the "Avoid" and "Shift" strategies, which incur nearly zero or low monetary costs, more investment costs are required for the "Improve" strategy because it is highly

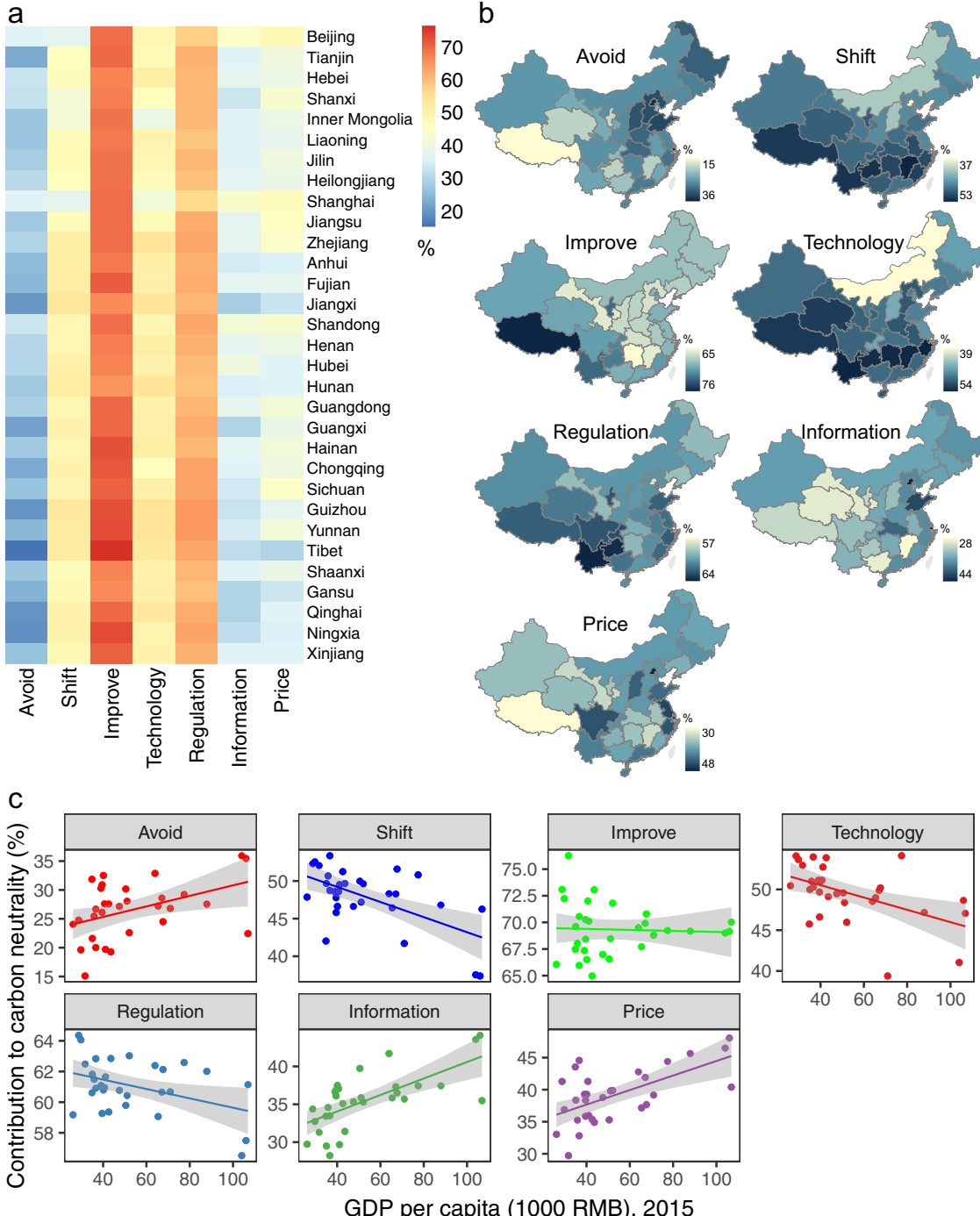

**Fig. 5 Regional differentiation of emission reductions. a** The heatmap indicates the contribution to the carbon neutrality target in 31 provinces under seven strategies and instruments. The horizontal axis is the three strategies and four instruments, while the vertical axis is the 31 provinces. **b** The geographical information system (GIS) map shows the geographic distributions of emission reduction potentials under seven strategies and instruments in 31 provinces. **c** The scatter diagram shows the correlation between GDP per capita and the contribution to carbon neutrality due to different strategies and instruments. The shaded areas in **c** show 95% confidence intervals.

technology-dependent and chiefly constrained to the national or regional economic development levels. Thus, if the "Improve" strategy that requires a large amount of technology investment can be matched with the "Avoid" and "Shift" strategies, the economic pressure for the introduction of new technologies may be eased by the reduction of transport demand and modal shift.

In addition to the synergies and trade-offs among strategies and instruments, it was observed that the impacts of the policy measures varied spatially across the provinces in China, implying

that the regional disparities in policy effectiveness require careful attention when making carbon-neutral transport policy and planning decisions. Actions taken as part of the "Avoid" strategy, such as land use control and transit-oriented development, produced more significantly positive effects in eastern developed regions because of the high urbanization levels and car dependence in developed provinces of China. This suggests that the feasibility and effectiveness of the transport management and spatial planning approaches are determined by the stage of

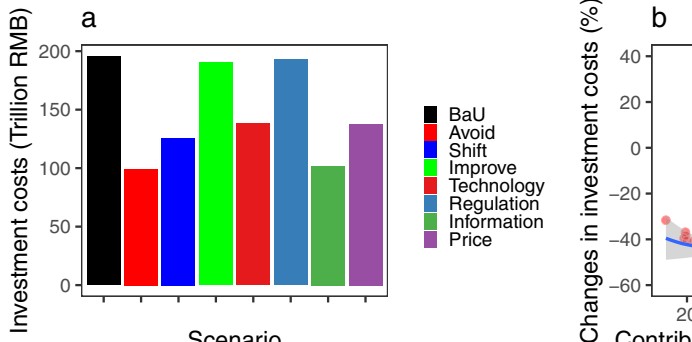
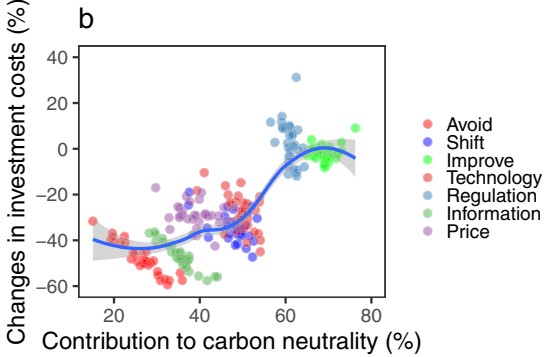

**Fig. 6 Economic costs under different strategies and instruments. a** Cumulative investment costs from 2015 to 2060 in 13 scenarios. **b** Correlation between the contribution to carbon neutrality and the investment costs in China's 31 provinces. The shaded areas in panel **b** show 95% confidence intervals.

economic development and urbanization in different regions. For actions taken as part of the "Improve" strategy, such as the adoption of EVs and renewable energy, the technological and economic feasibility of policy measures depends greatly on the development level of vehicle and energy technologies, which might differ across developed and developing provinces. The financial incentives for adopting advanced low-carbon technologies and renewable energy impose economic costs that will generate further pressure on economically underdeveloped provinces.

Therefore, the main findings of the scenario simulations shed light on synergistic coupling and trade-offs among different A-S-I strategies and instruments for prescribing a desirable mix of policy measures targeting the carbon neutrality target. To address long-term emissions reduction needs for China's transport sector, concrete policy recommendations must be presented to maximize the synergies and minimize the trade-offs among strategies and instruments. First, planning compact and dense cities is now the sustainable paradigm in urban planning. The key point that cannot be ignored is that the compact city should be planned together with a good public transport network. The introduction of teleworking and shared mobility could also enhance policy effectiveness of compact city policies. Second, the adoption of environmentally friendly vehicles needs to be promoted through various financial and non-financial incentives such as future bans on diesel cars, fuel taxes, purchase subsidies, vehicle rebates or vouchers, exemption from registration taxes, free public parking, and exemption from toll charges. Concurrent implementation of clean vehicle and ride-sharing policies can amplify the emission reduction effects of technological improvement policies. Also, a policy mix integrating clean vehicle deployment and behavioral inventions can mitigate the economic pressure from investment in advanced technologies. Third, there was a spatially differentiated policy effectiveness and feasibility when implementing the low-carbon policy measures, which were due to the different development stages, travel patterns, types of transport systems, vehicle technologies, and energy mixes across China's 31 provinces. Therefore, a synthetically designed policy package integrating the A-S-I strategies and corresponding to the regional disparities of the implementation effects of low-carbon policy measures is needed.

## Discussion
We investigated the long-term pathways and strategies toward a carbon-neutral ground transport sector in China by 2060, using a regional transport-energy integrated model that accommodates regional differences in economic development, demography, land use, transport infrastructure, transport cost, transport technological improvements, and policy interventions. The A–S–I

framework was employed to structure various transport strategies and instruments for identifying the effectiveness and feasibility of low-carbon policy measures. Because of the different disciplinary conceptions and methodologies, high-resolution behavioral and technological factors in the transport sector are difficult to assess quantitatively in current-generation IAMs. There have been few discussions of the positive role of the transport sector from an overall perspective of transport demand reduction, modal shift, and technology improvements in an integrated manner within the climate change mitigation agenda. To close the distance between transport and climate change studies, transport planners, energy policymakers, and climate experts need to come together to develop innovative solutions toward carbon neutrality.

There are a few limitations to this study that need to be acknowledged and addressed in future research. In the cost analysis of different A-S-I strategies and instruments, we only considered the financial cost of device investment, but the economic costs of behavioral interventions were excluded. For instance, the economic costs of behavioral transformation related scenarios are not truly reflected without considering behavioral costs. Meanwhile, the positive impacts on economic costs of technology improvement related scenarios would be overestimated because the device costs of investing in new low-carbon technologies would decrease with the reduction of transport demand and positive modal shift, but the behavioral costs that contributed to transport demand reduction and modal shift were not included. In addition, the decarbonization of the transport sector would be impossible without appropriate infrastructure such as electric quick-charging stations, but transport infrastructure costs were not included in the model, which needs more attention when improving the modal structure in the future.

While this study provides a comprehensive perspective on the carbon-neutral transport development based on the A-S-I approach, there is room for improvement in how a sustainable transport system should be conceptualized, modeled, assessed, and designed. In recent years, the A-S-I approach has been extended to Avoid-Shift-Share-Improve to explicitly consider the added value of shared transport such as ride-hailing, car sharing, micro-mobility (bike sharing and shared e-scooters), and on-demand micro-transit, because shared mobility services are increasingly introduced in transport systems and the sharing transport economy offers an opportunity to reduce the total number of vehicle-kilometers of travel and corresponding $CO_2$ emissions[25]. Although shared mobility was characterized as one of the "Shift" strategies in this paper, it should be highlighted as the "Share" solution in follow-up research.

Moreover, future research is required to further clarify the role of the transport sector in achieving carbon neutrality. This study only considered transport technologies that are currently in

operation or are likely to be popularized in the near future but did not account for some new technologies such as personal aerial vehicles, self-driven cars and trucks, package delivery drones, and delivery robots, which will not be applied reliably without a major technological breakthrough. The integrated transport energy model in this paper mainly involved China's ground transport sector that accounts for an overwhelming proportion of the total transport-related $CO_2$ emissions but did not include air and water transport due to the lack of reliable data at the regional level in China. Also, the scenarios were designed without considering the potential effects of the global coronavirus disease pandemic on economic growth and mobility, which warrants future attention if reliable data are available.

## Methods

**Transport model.** To represent the regional characteristics of transport dynamics rather than a national aggregated overview, we developed a regional transport model to offer spatially flexible and temporally dynamic simulations of transport demand and modal choices for China's ground transport sector in 31 provinces. For transport demand forecasting, panel data were used to predict the future transport demand for passenger and freight activities at the regional level. The data contained various socioeconomic driving factors, including GDP, population, transport infrastructure supply (e.g., road networks), land use, and generalized transport cost. The transport demand for passenger and freight transport in each region was calculated using Eq. (1):

$$TD_{i,n} = \eta_{i,n} \times GDPCAP_i^{\alpha_n} \times ROADCAP_i^{\beta_n} \times LANDCAP_i^{\gamma_n} \times COST_{i,n}^{\delta_n} \times POP_i \quad (1)$$

where $TD_{i,n}$ represents the total transport demand of passengers or freight in region $i$; $n$ denotes the transport sub-sectors, namely, passenger and freight; $GDPCAP_i$, $ROADCAP_i$, $LANDCAP_i$, and $COST_{i,n}$ represent GDP per capita, road length per capita, the area of built-up land per capita, and the generalized transport cost in region $i$, respectively; $POP_i$ represents the population in region $i$; and $\eta_{i,n}$, $\alpha_n$, $\beta_n$, $\gamma_n$, and $\delta_n$ are parameters to be estimated.

The transport demand was split into different modes considering detailed transport behavioral and technological descriptions such as mode preference, vehicle speed, travel time, load factor, device cost, and fuel cost. Seven modes of ground transport were considered in the transport model: car, bus, two-wheeler, and passenger rail for passenger transport and small truck, large truck, and freight rail for freight transport, respectively. The modal split was determined by choice modes in terms of the mode-specific cost. The probability of an available transport mode being selected was dependent on the transport cost of each mode, including the device, fuel, and time costs. The mode-specific transport cost $MCOST_{i,n,m}$ and modal share $SHARE_{i,n,m}$ were calculated using Eqs. (2) and (3):

$$MCOST_{i,n,m} = DCOST_{i,n,m} + FCOST_{i,n,m} + TCOST_{i,n,m} \quad (2)$$

$$SHARE_{i,n,m} = \frac{\theta_{i,n,m} \times MCOST_{i,n,m}^{\sigma_n}}{\sum_m \theta_{i,n,m} \times MCOST_{i,n,m}^{\sigma_n}} \quad (3)$$

where $DCOST_{i,n,m}$, $FCOST_{i,n,m}$, and $TCOST_{i,n,m}$ represent the device, fuel, and time costs of mode $m$ for a unit of transport demand in region $i$, respectively; and $\theta_{i,n,m}$ and $\sigma_n$ are parameters of the modal choice function.

*Model coupling with an energy system model.* The transport model was coupled with an energy system model to determine the optimal technology and energy mix, and the corresponding emissions in the transport sector (see Supplementary Fig. 1). A bottom-up technology detailed recursive dynamic optimization model, the Asia-Pacific Integrated Model (AIM)/Enduse, was used to describe the energy system[26]. The optimal set of technologies at the minimum total system cost was selected by AIM/Enduse based on a detailed technology database, which includes information on the lifetime, initial costs, operation and maintenance costs, energy efficiency, and emission factors. To satisfy a given amount of transport demand, technology selection was modeled as a linear optimization problem with some constraints to minimize the total system costs. A variety of technologies, including existing and emerging technologies for China's ground transport sector and a detailed list of techniques considered in the energy system model, are provided in Supplementary Table 1.

The transport model passed the mode-specific demand to the energy system model to estimate the future technology mix, energy consumption, and $CO_2$ emissions, while the technology mix and costs were fed back to the transport model to re-calculate the generalized transport cost with the updated technology mix. An iterative procedure was used to achieve the convergence of the coupled transport-energy system model. This feedback loop continued until a self-consistent solution was obtained, in which the technology mix and generalized transport costs remained constant in successive iterations. The study considered a time span of 45 years from 2015 to 2060 at 1-year intervals. The integrated transport energy model was developed with the General Algebraic Modeling System (GAMS) 33.2.0.

**Study area and data.** This study considered all 31 provinces, autonomies, and municipalities of mainland China, excluding Hong Kong, Macau, and Taiwan due to inconsistencies in statistical standards (Supplementary Figs. 2, 3). Parameters for the transport model were estimated based on historical data at China's provincial level from 2002 to 2015. Socioeconomic, land use, and infrastructure data, including GDP, population, area of built-up land, and road length, were acquired from the Chinese Statistical Yearbook (2002–2015). Historical transport demand data for road transport modes in terms of person-km and ton-km were calculated based on the number of vehicles in use reported in the Chinese Statistical Yearbook (2002–2015), while vehicle-km travelled and load factors were obtained from existing studies[27,28]. Passenger and freight rail demands were acquired directly from the Yearbook of China Transportation & Communications (2002–2015). The device, fuel, and time costs were used to determine the transport costs and were calculated using historical fuel prices, energy intensity, and door-to-door travel speeds collected from existing studies[29–31]. For the bottom-up technology detailed recursive dynamic optimization model, we established a detailed technology database, including the initial cost, lifetime, energy efficiency, and emission factors, based on the AIM/Enduse datasets and related studies[32,33].

Historical panel data on GDP, population, area of built-up land, road length, generalized transport cost, and transport demand from 2002 to 2015 were used to estimate the parameters of the panel data models used to project the future passenger and freight transport demand. We employed three types of panel analytic models: (1) pooled regression model, (2) fixed effect (FE) model, and (3) random effect (RE) model to estimate the parameters $\alpha_n$, $\beta_n$, $\gamma_n$, and $\delta_n$ for passenger and freight transport, respectively. The results of the pooled regression model, FE model, and RE model were compared using the F-test and Hausman test to statistically determine the most appropriate model (see Supplementary Table 2). The cost distribution parameter $\sigma_n$ in the modal choice function is borrowed from previous literature[31,34]. Parameters $\eta_{i,n}$ and $\theta_{i,n,m}$ in the transport demand projection and modal choice functions were calibrated using the baseline year data in 2015 to match observed data of transport demand and modal shares with simulation results in 2015. Model parameter estimates of the transport model are provided in the supplementary information.

Socioeconomic underlying factors for future projections, such as GDP and population pathways from 2016 to 2060, were obtained by downscaling the Shared Socioeconomic Pathways 2 database from the national to provincial levels. The Shared Socioeconomic Pathways 2 narrative describes a middle-of-the-road scenario, which depicts China's population peaking at approximately 1.4 billion by the end of the current decade[35]. Data for future land use and infrastructure, such as the area of built-up land and road length, were established based on land use and infrastructure planning in China[36].

**Scenario settings.** To investigate the low-carbon transport actions toward carbon neutrality from the perspectives of transport demand management, modal structure, and technology improvement, a set of scenarios was created to project the long-term (up to 2060) trends of transport demand, modal choices, energy use, and emission profiles according to the A-S-I framework. The A-S-I matrix classifies the low-carbon policy measures for decarbonizing the transport sector into two dimensions. One is the strategy, including "Avoid", "Shift", and "Improve", while the other is instruments, including "Technology", "Regulation", "Information", and "Price". We therefore created many complex scenarios for possible low-carbon policy measures. These scenarios were defined according to transport demand management, modal shift, and technology improvements, which were determined by the instruments of technology, regulation, information, and price and by varying the assumptions and parameters in the integrated transport energy model.

Under the "Avoid" strategy, four scenarios, A_Tech, A_Regu, A_Info, and A_Pric, were created to describe the technology, regulation, information, and price instruments, respectively. In the A_Tech scenario, transit-oriented development (TOD) was considered the typical technological instrument, with the assumption that the TOD design can reduce both the length and quantity of driving trips. The parameter $\eta$ in the transport model was assumed to represent the impact of the urban development mode in isolation from other socioeconomic factors, such as GDP, population, land use, infrastructure, and transport cost. For all provinces, this factor gradually converged to Chongqing's 2015 value by 2060. Because Chongqing had the lowest value and was considered a typical area of the TOD mode. In the A_Regu scenario, we considered the creation of compact cities as a land use planning tool to represent the regulatory instrument. Due to the urban compactness and high-density land use development, the addition of new built-up areas will decrease by 20% by 2060[37]. In the A_Info scenario, the informatic instruments could be interpreted as the prevalence of teleworking, which reduced the future transport demand. The parameters in the transport model that represented teleworking were calibrated by the empirical transport data in China's 31 provinces in 2020, with the implementation of teleworking under the physical distancing measures and lockdown in the COVID-19 pandemic. In the A_Pric scenario, road pricing for cars and trucks was a specific economic policy measure to reduce the transport demand. The target value for road pricing in 2060 was set by referring to the example of the congestion charge in London (£11.50 per day).

The "Shift" strategy also included four scenarios: S_Tech, S_Regu, S_Info, and S_Pric as instruments of technology, regulation, information, and price, respectively. In the S_Tech scenario, the continuously expanded construction of the

high-speed railway network in China was used as a technological instrument to describe the potential modal shift from road transport to rail transport. The mode preference factors of all provinces in the transport model by 2060 gradually converged to Jiangxi's preference factor for passenger transport and Inner Mongolia's preference factor for freight transport in 2015, respectively. In the S_Regu scenario, regulatory instruments such as the promotion of public transport (e.g., prioritization of buses) reflected the modal shift from private to public transport in the model. The mode preference factors of all provinces in the transport model by 2060 gradually converged to Shanghai's preference factor in 2015. In the S_Info scenario, a wireless communication system for promoting carpooling and car-sharing services would give customers the option to shift from an energy-consuming mode to an energy-saving mode due to the higher occupancy rates of vehicles. This scenario assumed that the load factors of cars, buses, and trucks would increase by 50% by 2060[16]. In the S_Pric scenario, a fuel tax for fossil fuels was imposed as an economic instrument to promote the mode shift strategy. The fuel tax rate in China is about 30%, while the fuel tax rates in developed countries such as Germany, the Netherlands, France, and Norway are as high as 100–200%. In this scenario, we assumed that China's future fuel tax rate would also increase significantly, leading to the price of gasoline and diesel doubling by 2060.

In the "Improve" strategy, four scenarios were structured for the instruments of technology (I_Tech), regulation (I_Regu), information (I_Info), and price (I_Pric). In the I_Tech scenario, the technological development of renewable energy was regarded as the potential technological instrument. We assumed a 50% biofuel blend ratio (B50) in transport fuel by 2060. In the I_Regu scenario, the internal combustion engine car was banned, and the stringent penetration of EVs served as the key regulatory instrument for decarbonizing the transport sector. A 100% market share of battery EVs for cars, buses, two-wheelers, and small trucks was assumed for 2060, and it was also anticipated that all large trucks would be FCVs[20,38,39]. In the I_Info scenario, an intelligent transport system and eco-driving patterns helped reduce fuel consumption and $CO_2$ emissions. This scenario assumed that the energy efficiency of cars and trucks would increase by 5% due to eco-driving patterns[40]. In the I_Pric scenario, subsidies for EVs and FCVs were offered as an economic instrument based on the subsidy standard in China.

In addition to each individual scenario designed based on the A-S-I approach, a series of combined scenarios were created to identify the emission reduction effects of different strategies and instruments. All instruments under the same strategy were combined for the scenarios of "Avoid", "Shift", and "Improve", while the same instruments under different strategies were grouped into the combined scenarios of "Technology", "Regulation", "Information", and "Price". In these combined scenarios, all assumptions and parameters under the same strategy or instrument category were given simultaneously. Furthermore, a BaU scenario, which presumed the continuation of technological improvements at the current pace and maintenance of the existing transport and energy policies, was designed to explore the default pathways of energy transition and emission reduction in China's transport sector. In particular, the setting of technological development in the BaU scenario is mainly based on the data and assumptions made in China's Development Plan for the New Energy Automobile Industry. By comparing the scenarios that assume exogenous monetary policy shocks with the BaU scenario, the impacts of potential low-carbon options would be investigated to propose a synthetically designed policy package toward carbon neutrality by the mid-21st century.

**Reporting summary**. Further information on research design is available in the Nature Research Reporting Summary linked to this article.

## Data availability

Model output data are available at https://doi.org/10.5281/zenodo.6617098. The Shared Socioeconomic Pathways 2 database used for this model is available at https://tntcat.iiasa.ac.at/SspDb/dsd?Action=htmlpage&page=about. The source data underlying Figs. 2–6 are provided with this paper.

## Code availability

All code used for data analysis and creating the figures is available via Zendo at https://zenodo.org/record/6617098#.Yp4OVqhByF4.

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

## Acknowledgements

R.Z. acknowledges financial support from the Japan Society for the Promotion of Science (JSPS) (Grant 21K17926, 21KK0208). R.Z. and T.H. were supported by the Environment Research and Technology Development Fund S-20-3 (JPMEERF21S12009) of the Environmental Restoration and Conservation Agency of Japan. T.H. received funding from the Environment Research and Technology Development Fund 2-1908 (JPMEERF20192008) of the Environmental Restoration and Conservation Agency of Japan.

## Author contributions

R.Z. and T.H. conceived and designed the conceptual framework for this study. R.Z. performed calculations, simulations, data analysis, and wrote the paper. T.H. discussed the results and revised the manuscript.

## Competing interests

The authors declare no competing interests.
