## [Peer Review File · Nature Communications]

Reviewer comments, first round review

Reviewer #1 (Remarks to the Author):

I would like to congratulate the authors on a highly stimulating and informative research study. The research is very timely and is highly important for informing policy decisions and strategies for effective and sustainable pathways to decarbonisation of the transport sector focused on China. The methodologies used are highly appropriate for the scale of the study and the results obtained are consistent with what has been reported and expected in the wider literature.

The paper in its current form strikes a good balance between providing the necessary background information, methodologies and results. However, I feel that the article can benefit from a number of amendments as described below.

1. First, the paper needs a restructure of the sections. For example, the authors present the results and findings before they describe the modelling approach and methodologies. I suggest that it should be the other way around. First, provide the methodology and clarify the scenarios being tested, then present the results.

2. Second, the paper is missing proper delineation between the different sections. For example, the first page starts with the abstract but then it goes into the introduction but there is no section heading for the introduction. I suggest that section headings need to be provided to improve readability

3. In the methodology, the authors consider the well-known Avoid, Shift, Improve strategy. However, in recent times, this approach has been extended to "Avoid, Shift, Share, Improve" to take into consideration the added value of shared transport such as ride-hailing and car-sharing, micro-mobility shared e-bikes and e-scooters etc, UBER, DIDI and similar solutions. While the authors have included some of these in the analysis, it would improve the paper if it is aligned with the newer thinking on the topic. The authors can review a book that was published in 2017 and the relevant chapters on the topic:

<https://digital-library.theiet.org/content/books/tr/pbtr006e>

4. A conclusions section is missing. The authors have reported on very valuable scenarios and obtained great results, it is therefore important to provide a summary or conclusions section at the end summarising to the reader the overall findings and value of the work.

Reviewer #2 (Remarks to the Author):

The manuscript titled "Cross-cutting scenarios and strategies for designing decarbonization pathways in China's transport sector toward carbon neutrality presents a novel, well-documented

research. The most important innovation is that the work provided scenarios based on an integrated framework that accommodates all possible strategies (Avoid, Shift, and Improve) and instruments (Technology, Regulation, Information, and Economy). However, I find several major concerns that I would want the authors to fully address before the work being considered for publication. This may require the authors to re-write most part of the manuscript.

First, I would suggest that the authors clearly articulate the method and improve academic rigor throughout. Projecting the future development of China's transport sector for carbon neutrality requires (i) estimating regional preference parameters for transport demands and modal choices and reflecting them to the integrated framework to generate scenarios and (ii) borrowing parameters or numbers from previous literature for those unobserved historically. But, the manuscript is not very clear about the process of estimating the preference parameters (e.g., sample size, significance, distribution of estimates, etc.) and also does not provide any academic references for the process (e.g., equations 1 and 2). Another example of missing reference is for future land use and infrastructure information.

Parameterization of strategy-instrument scenarios does not seem rigorous either in the method. They assume the addition of new built-up areas decreasing by 20% by 2060 but without any reference. The target value for road pricing in 2060 is from the congestion charge in London; but what exact level? Load factors are assumed to increase by 50% by 2060; was it substantiated by any previous study? The price of gasoline and diesel doubled by 2060, how come? The proportion of vehicles running on biofuel increased to 50% by 2060; is this valid assumption? 100% market share of clean vehicles by 2060; is this assumption reasonable? Energy efficiency of cars and trucks increase by 5% due to eco-driving; is this shown by any reference?

Second, I was surprised to see economic costs in several cases decreasing substantially with policy intervention! Maybe the authors are not reflecting direct costs of implementing behavioral interventions and information programs on top of the BAU cost? Or perhaps the BAU world is operating in a suboptimal way, and some of the interventions moves the equilibrium more to the optimal level? But this contradicts their explanation that the energy system model solves a linear optimization problem with some constraints. The authors should be clearer about the process of calculating economic costs and provide convincing explanation about economic costs.

The third point relates to the first two. Without providing much more confidence in research methods, I think it may be too early to say that the improvement strategy would be the most effective way of reducing carbon dioxide emissions from transport and that considerable investment costs would be required to implement the improvement strategy than for the avoid and shift strategies.

Forth, the authors can do more to present concrete policy recommendations than suggesting that the improvement strategy is most effective and a region-specific policy package is needed. Perhaps the work can explore opportunities for synergistic coupling (or trade-offs) between strategies and instruments for carbon neutrality. Or it can prescribe a desirable mix of strategies and instruments for individual Chinese provinces.

Minor comments: (1) The authors may want to motivate readers in the introduction part by highlighting the importance of the transport sector in China's national GHG emissions. (2) I would suggest "price" instead of economy as one of the four instruments (3) energy efficiency should not

be confused with energy intensity (of service). See lines 188 and 192 on page 8 and axes in Figure 4.
(4) How do we know "policy effectiveness might be due to economic development at the local level"? (line 230 on page 10) Provide evidence.

Reviewer #3 (Remarks to the Author):

This paper is intended to depict the long-term pathways to deep decarbonization of China's transport sector by using a regional transport-energy integrated model. Overall, the methodology and scenario simulations based on the Avoid-Shift-Improve framework are well elaborated and, as a whole, easy to follow, particularly with growing interest in the deep decarbonization of transport sector. I outline some specific concerns below.

(1) Regarding the CO₂ emission pathways in Fig.3, it would be better to provide more details on how to set the combinations of 12 scenarios.

(2) In Fig. 4, the CO₂ emissions decreased significantly from 2015 to 2060 in the BaU scenario without implementing any Avoid-Shift-Improve policies. I think it should be more clearly indicated how the storyline of BaU scenario was constructed for this study.

(3) Since Fig. 5 contains multi-panels, it is recommended to add panel titles like a, b, c for heatmap, GIS map, and scatter diagram.

(4) When analyzing the regional disparities in policy impacts, it is not clear to an international reader where the provinces are located in China and why some of the provinces can be described as developing or developed. Can the authors provide an additional information and metric for developing and developed regions?

(5) Regarding the economic costs in Fig. 6, I think it needs to be indicated somewhere whether the infrastructure cost has been included.

(6) The pandemic will exert a profound effect on the decarbonization of China's transport sector. Did the authors consider the impacts of the pandemic on GDP, population, mobility, EV production, and behaviors in this study?

(7) With respect to the methodology, it would be better to provide a diagram for more details on how the model works, and particularly how the transport model was coupled with an energy system model, at least in the supplementary information.

(8) Why did you not consider air and water transportation? They consume a lot of liquid fossil fuels and generate a massive amount of carbon emissions.

(9) How about the advanced technologies such as personal aerial vehicles, package delivery drones, robots for delivery? Will the transport sector be influenced by the technological innovations in the future?

Point-by-point responses:

Reviewer 1's general comments

I would like to congratulate the authors on a highly stimulating and informative research study. The research is very timely and is highly important for informing policy decisions and strategies for effective and sustainable pathways to decarbonisation of the transport sector focused on China. The methodologies used are highly appropriate for the scale of the study and the results obtained are consistent with what has been reported and expected in the wider literature.

The paper in its current form strikes a good balance between providing the necessary background information, methodologies and results. However, I feel that the article can benefit from a number of amendments as described below.

Response: We are grateful to the reviewer for your appreciation of the value of our work and we sincerely thank the reviewer for these insightful and valuable suggestions and comments. The revisions and responses to specific comments are shown below.

Reviewer 1's specific comments 1)

1. First, the paper needs a restructure of the sections. For example, the authors present the results and findings before they describe the modelling approach and methodologies. I suggest that it should be the other way around. First, provide the methodology and clarify the scenarios being tested, then present the results.

Response: Thank you very much for pointing up this issue. According to the format requirements of the journal, the methodology needs to be listed as a separate Methods section after the main text. The current structure of the paper into sections is in compliance with the journal requirements. We agree with the reviewer's comment that the logical structure of the paper is important for the reader to clearly capture the point and content of our paper. Therefore, we have highlighted the description and explanation of the method in the final paragraph of the Introduction section.

Reviewer 1's specific comments 2)

2. Second, the paper is missing proper delineation between the different sections. For example, the first page starts with the abstract but then it goes into the introduction but there is no section heading for the introduction. I suggest that section headings need to be provided to improve readability

Response: Thank you very much for mentioning this concern. We have added a section heading for the introduction to improve readability.

Reviewer 1's specific comments 3)

3. In the methodology, the authors consider the well-known Avoid, Shift, Improve strategy. However, in recent times, this approach has been extended to "Avoid, Shift, Share,

Improve" to take into consideration the added value of shared transport such as ride-hailing and car-sharing, micro-mobility shared e-bikes and e-scooters etc, UBER, DIDI and similar solutions. While the authors have included some of these in the analysis, it would improve the paper if it is aligned with the newer thinking on the topic. The authors can review a book that was published in 2017 and the relevant chapters on the topic:

<https://digital-library.theiet.org/content/books/tr/pbtr006e>

Response: Thank you very much for providing the helpful references. We have carefully read this book in detail and added it to the reference list of this paper. As suggested by the reviewer, we agree that the "Avoid, Shift, Share, Improve" framework is worthy of study, since the "Avoid, Shift, Improve" approach has been extended in consideration of the shared mobility. According to this comment, the following text has been added to the main text to show the limitations of the current "Avoid, Shift, Improve" framework and the need to extend to the "Avoid, Shift, Share, Improve" framework for future studies.

"While this study provides a comprehensive perspective on the carbon-neutral transport development based on the A-S-I approach, there is room for improvement in how a sustainable transport system should be conceptualized, modeled, assessed, and designed. In recent years, the A-S-I approach has been extended to Avoid-Shift-Share-Improve to explicitly consider the added value of shared transport such as ride-hailing, car sharing, micro-mobility (bike sharing and shared e-scooters), and on-demand micro-transit, because shared mobility services are increasingly introduced in transport systems and the sharing transport economy offers an opportunity to reduce the total number of vehicle-kilometers of travel and corresponding CO₂ emissions (Dia, 2017). Although shared mobility was characterized as one of the "Shift" strategies in this paper, it should be highlighted as the "Share" solution in follow-up research."

Reviewer 1's specific comments 4)

4. A conclusions section is missing. The authors have reported on very valuable scenarios and obtained great results, it is therefore important to provide a summary or conclusions section at the end summarising to the reader the overall findings and value of the work.

Response: Thank you very much for your comments. We have added a separate conclusion section to provide a summary of the overall findings, the value of the work, and limitations, as follows:

"We investigated the long-term pathways and strategies toward a carbon-neutral ground transport sector in China by 2060, using a regional transport-energy integrated model that accommodates regional differences in economic development, demography, land use, transport infrastructure, transport cost, transport technological improvements, and policy interventions. The A-S-I framework was employed to structure various transport strategies and instruments for identifying the effectiveness and feasibility of low-carbon policy measures. Because of the different disciplinary conceptions and methodologies,

high-resolution behavioral and technological factors in the transport sector are difficult to assess quantitatively in current-generation IAMs. There have been few discussions of the positive role of the transport sector from an overall perspective of transport demand reduction, modal shift, and technology improvements in an integrated manner within the climate change mitigation agenda. To close the distance between transport and climate change studies, transport planners, energy policymakers, and climate experts need to come together to develop innovative solutions toward carbon neutrality.

There are a few limitations to this study that need to be acknowledged and addressed in future research. In the cost analysis of different A-S-I strategies and instruments, we only considered the financial cost of device investment, but the economic costs of behavioral interventions were excluded. For instance, the economic costs of behavioral transformation related scenarios are not truly reflected without considering behavioral costs. Meanwhile, the positive impacts on economic costs of technology improvement related scenarios would be overestimated because the financial costs of investing in new low-carbon technologies and new devices would decrease with the reduction of transport demand and positive modal shift, but the economic costs of behavioral interventions that contributed to transport demand reduction and modal shift were not included. In addition, the decarbonization of the transport sector would be impossible without appropriate infrastructure such as electric quick-charging stations, but transport infrastructure costs were not included in the model, which needs more attention when improving the modal structure in the future.

While this study provides a comprehensive perspective on the carbon-neutral transport development based on the A-S-I approach, there is room for improvement in how a sustainable transport system should be conceptualized, modeled, assessed, and designed. In recent years, the A-S-I approach has been extended to Avoid-Shift-Share-Improve to explicitly consider the added value of shared transport such as ride-hailing, car sharing, micro-mobility (bike sharing and shared e-scooters), and on-demand micro-transit, because shared mobility services are increasingly introduced in transport systems and the sharing transport economy offers an opportunity to reduce the total number of vehicle-kilometers of travel and corresponding CO₂ emissions. Although shared mobility was characterized as one of the “Shift” strategies in this paper, it should be highlighted as the “Share” solution in follow-up research.

Moreover, future research is required to further clarify the role of the transport sector in achieving carbon neutrality. This study only considered transport technologies that are currently in operation or are likely to be popularized in the near future but did not account for some new technologies such as personal aerial vehicles, self-driven cars and trucks, package delivery drones, and delivery robots, which will not be applied reliably without a major technological breakthrough. The integrated transport energy model in this paper mainly involved China’s ground transport sector that accounts for an overwhelming

proportion of the total transport-related CO₂ emissions but did not include air and water transport due to the lack of reliable data at the regional level in China. Also, the scenarios were designed without considering the potential effects of the global coronavirus disease pandemic on economic growth and mobility, which warrants future attention if reliable data are available.”

Reviewer 2's general comments

The manuscript titled "Cross-cutting scenarios and strategies for designing decarbonization pathways in China's transport sector toward carbon neutrality presents a novel, well-documented research. The most important innovation is that the work provided scenarios based on an integrated framework that accommodates all possible strategies (Avoid, Shift, and Improve) and instruments (Technology, Regulation, Information, and Economy). However, I find several major concerns that I would want the authors to fully address before the work being considered for publication. This may require the authors to re-write most part of the manuscript.

Response: We sincerely thank the reviewer for constructive criticisms and valuable comments, which were of great help in revising the manuscript. The insightful and valuable comments allowed us to realize that we should improve the clarity of the paper for readers by adding more information and clarifications. We therefore rewrote most parts of this manuscript, particularly the methodology, synergistic coupling and trade-offs among strategies and instruments, and concrete policy recommendations. Please find the point-to-point responses to the specific comments listed below.

Reviewer 2's specific comments 1)

First, I would suggest that the authors clearly articulate the method and improve academic rigor throughout. Projecting the future development of China's transport sector for carbon neutrality requires (i) estimating regional preference parameters for transport demands and modal choices and reflecting them to the integrated framework to generate scenarios and (ii) borrowing parameters or numbers from previous literature for those unobserved historically. But, the manuscript is not very clear about the process of estimating the preference parameters (e.g., sample size, significance, distribution of estimates, etc.) and also does not provide any academic references for the process (e.g., equations 1 and 2). Another example of missing reference is for future land use and infrastructure information.

Parameterization of strategy-instrument scenarios does not seem rigorous either in the method. They assume the addition of new built-up areas decreasing by 20% by 2060 but without any reference. The target value for road pricing in 2060 is from the congestion charge in London; but what exact level? Load factors are assumed to increase by 50% by 2060; was it substantiated by any previous study? The price of gasoline and diesel doubled by 2060, how come? The proportion of vehicles running on biofuel increased to 50% by 2060; is this valid assumption? 100% market share of clean vehicles by 2060; is

this assumption reasonable? Energy efficiency of cars and trucks increase by 5% due to eco-driving; is this shown by any reference?

Response: Thank you very much for highlighting this issue. We agree with the reviewer's option that the method should be clearly articulated to improve academic rigor throughout. Following this comment, we have provided the process and results of estimating the parameters for transport demands and modal choices and the parameter estimation results in the supplementary information (Supplementary Table 2). We have also added more text in the Methods section to identify the process and results of parameter estimation:

“We employed three types of panel analytic models: (1) pooled regression model, (2) fixed effect (FE) model, and (3) random effect (RE) model to estimate the parameters α_n , β_n , γ_n , and δ_n for passenger and freight transport, respectively. The results of the pooled regression model, FE model, and RE model were compared using the F-test and Hausman test to statistically determine the most appropriate model (see Supplementary Table 2). The cost distribution parameter σ_n in the modal choice function is borrowed from previous literature. Parameters $\eta_{i,n}$ and $\theta_{i,n,m}$ in the transport demand projection and modal choice functions were calibrated using the baseline year data in 2015 to match observed data of transport demand and modal shares with simulation results in 2015.”

With respect to the future land use and infrastructure information, we have added the references for China's land use and infrastructure planning in the future.

Regarding the parameterization of strategy-instrument scenarios, we have added more explanations on parameter settings and the references in each scenario. For the new built-up areas in the A_Regu scenario, we assumed that the addition of new built-up areas would decrease by 20% by 2060. According to the “13th Five-Year Plan of Land and Resources” issued by the Ministry of Land and Resources, the total amount of construction land should be effectively controlled, and the construction land use area per unit of GDP will be reduced by 20%. We have added this reference for the A_Regu scenario in the Methods section.

For the road pricing setting in the A_Pric scenario, we have added the exact level of the congestion charge in London as follows:

“The target value for road pricing in 2060 was set by referring to the example of the congestion charge in London (£11.50 per day).”

For the load factor in the S_Info scenario, we have added the reference for setting the load factors in the future. According to previous studies, it was assumed that vehicle occupancy is estimated at a maximum of 2.5 persons per vehicle in developing countries.

Regarding the parameter setting in the S_Pric scenario, a fuel tax that refers to the tax

levied on fossil fuels, was assumed as an economic instrument to promote the mode shift strategy. The fuel tax rate in China is about 30%, while the fuel tax rates in developed countries such as Germany, the Netherlands, France, and Norway are as high as 100-200%. In this scenario, we assumed that China's future fuel tax rate would also increase significantly, leading to the price of gasoline and diesel doubling by 2060. We have revised the relevant statement in this section to clarify the parameter setting for the S_Pric scenario:

“The fuel tax rate in China is about 30%, while the fuel tax rates in developed countries such as Germany, the Netherlands, France, and Norway are as high as 100-200%. In this scenario, we assumed that China's future fuel tax rate would also increase significantly, leading to the price of gasoline and diesel doubling by 2060.”

For the I_Tech scenario, we apologize for the language errors. We assumed a 50% biofuel blend ratio (B50) in transport fuel by 2060 and we have corrected this sentence as follows:

“We assumed a 50% biofuel blend ratio (B50) in transport fuel by 2060.”

For the I_Regu scenario, a 100% market share of electric vehicles was assumed by 2060. According to China's Development Plan for the New Energy Automobile Industry (2021-2035), electric vehicles could reach a 25% market share by 2025, while the Chinese government has not released specific long-term plans for deploying electric vehicles by 2060. However, from a global perspective, a number of countries have set targets and timelines to phase out internal combustion engine vehicles by the mid-21st century. Therefore, we adopted this ambitious scenario to detect the impacts of stringent EV penetration according to global policy trends on EV deployment. There are a few pieces of literature that investigated the impact of the 100% electric vehicle policy, so we have added relevant literature here for this scenario.

For the I_Info scenario, relevant studies have revealed that the energy efficiency of cars and trucks would increase by 5% due to eco-driving patterns. We have added references for the eco-driving scenario.

Reviewer 2's specific comments 2)

Second, I was surprised to see economic costs in several cases decreasing substantially with policy intervention! Maybe the authors are not reflecting direct costs of implementing behavioral interventions and information programs on top of the BAU cost? Or perhaps the BAU world is operating in a suboptimal way, and some of the interventions moves the equilibrium more to the optimal level? But this contradicts their explanation that the energy system model solves a linear optimization problem with some constraints. The authors should be clearer about the process of calculating economic costs

and provide convincing explanation about economic costs.

Response: Thank you very much for pointing up this problem. For the cost analysis in this paper, we only considered the financial cost of device investment, but the transaction costs and economic costs of behavioral interventions were excluded. Various policy measures had significant impacts on reducing transport service demand and positively changing the modal structure. Thus, the financial costs of investing in new low-carbon technologies and devices would fall with the reduction of transport demand and positive modal shift. This is the reason why the economic costs decreased substantially with policy intervention. If the costs of behavioral interventions were considered in the cost analysis, the results may be affected to some extent. We apologize for the unclear explanation regarding the economic cost analysis. We fully agree with the reviewer's statement that while behavioral interventions are often not included in the cost of investments, their economic impacts need to be considered. We, therefore, have added an explanation illustrating the limitations of not accounting for the economic costs of behavioral interventions, and have indicated the direction of future research. The newly added text is as follows:

“There are a few limitations to this study that need to be acknowledged and addressed in future research. In the cost analysis of different A-S-I strategies and instruments, we only considered the financial cost of device investment, but the economic costs of behavioral interventions were excluded. For instance, the economic costs of behavioral transformation related scenarios are not truly reflected without considering behavioral costs. Meanwhile, the positive impacts on economic costs of technology improvement related scenarios would be overestimated because the financial costs of investing in new low-carbon technologies and new devices would decrease with the reduction of transport demand and positive modal shift, but the economic costs of behavioral interventions that contributed to transport demand reduction and modal shift were not included. In addition, the decarbonization of the transport sector would be impossible without appropriate infrastructure such as electric quick-charging stations, but transport infrastructure costs were not included in the model, which needs more attention when improving the modal structure in the future.”

Reviewer 2's specific comments 3)

The third point relates to the first two. Without providing much more confidence in research methods, I think it may be too early to say that the improvement strategy would be the most effective way of reducing carbon dioxide emissions from transport and that considerable investment costs would be required to implement the improvement strategy than for the avoid and shift strategies.

Response: Thank you very much for your comments. Following this comment, we have expanded the methodology description in the Methods section, particularly the parameterization and the model structure. We agree that it may be premature and arbitrary

to conclude that the improvement strategy will be the most effective way to reduce CO2 emissions from transport, and considerable investment costs would be required to implement the “Improve” strategy compared to the “Avoid” and “Shift” strategies. Therefore, in addition to clarifying our methodology, we have added two paragraphs to highlight that while it is beneficial to clarify the role of the “Improve” strategy, we need to recognize the synergies and trade-offs among strategies and instruments.

“Our findings neither depreciated the positive effects of transport demand management nor unilaterally overstated the contribution of technological improvements to the decarbonization of the transport sector. We rather highlighted a balanced perspective to detect the trade-offs among strategies and instruments as they impact upon the extent to which the ultimate intended goals or outcomes. Although the “Improve” strategy provides a plausible solution due to its most substantial positive effects on emission reductions, the improvement of fuel economy due to the “Improve” strategy was inclined to generate more travel demand, which may negatively affect the mitigation effects, particularly the demand reduction effects contributed by the “Avoid” strategy. Furthermore, policy effectiveness of the “Shift” strategy that encourages travelers to shift from private motorized vehicles to public transport may be weakened to some extent by the “Improve” strategy. For example, the petrol car ban and EV subsidies would increase the private car travel mode choice, which may negatively influence the use of public transport promoted by the policies prioritizing high-speed rail and buses.

While the trade-offs describe potential conflicts among different A-S-I strategies and instruments, policymakers must consider the synergistic coupling of various policy measures. The “Avoid” strategy, e.g., land use planning and compact city form, can reduce travel demand by cutting down on long-distance trips but might also generate negative externalities (e.g., traffic congestion in the city center), which would offset the benefits produced by the “Avoid” strategy. Because the externalities of congestion could probably be alleviated through promotion of the public transport, the “Avoid” strategy needs to be implemented with the adoption of policy instruments designed according to the “Shift” strategy. In addition, compared to the behavioral interventions of the “Avoid” and “Shift” strategies, which incur nearly zero or low monetary costs, more investment costs are required for the “Improve” strategy because it is highly technology-dependent and chiefly constrained to the national or regional economic development levels. Thus, if the “Improve” strategy that requires a large amount of technology investment can be matched with the “Avoid” and “Shift” strategies, the economic pressure for the introduction of new technologies may be eased by the reduction of transport demand and modal shift.”

Reviewer 2’s specific comments 4)

Forth, the authors can do more to present concrete policy recommendations than

suggesting that the improvement strategy is most effective and a region-specific policy package is needed. Perhaps the work can explore opportunities for synergistic coupling (or trade-offs) between strategies and instruments for carbon neutrality. Or it can prescribe a desirable mix of strategies and instruments for individual Chinese provinces.

Response: Thank you very much for your valuable advice. We have added a paragraph to present concrete policy recommendations to highlight the synergistic coupling and trade-offs among strategies and instruments for a desirable mix of strategies and instruments for individual Chinese provinces.

“Therefore, the main findings of the scenario simulations shed light on synergistic coupling and trade-offs among different A-S-I strategies and instruments for prescribing a desirable mix of policy measures targeting the carbon neutrality target. To address long-term emissions reduction needs for China’s transport sector, concrete policy recommendations must be presented to maximize the synergies and minimize the trade-offs among strategies and instruments. First, planning compact and dense cities is now the sustainable paradigm in urban planning. The key point that cannot be ignored is that the compact city should be planned together with a good public transport network. The introduction of teleworking and shared mobility could also enhance policy effectiveness of compact city policies. Second, the adoption of environmentally friendly vehicles needs to be promoted through various financial and non-financial incentives such as future bans on diesel cars, fuel taxes, purchase subsidies, vehicle rebates or vouchers, exemption from registration taxes, free public parking, and exemption from toll charges. Concurrent implementation of clean vehicle and ride-sharing policies can amplify the emission reduction effects of technological improvement policies. Also, a policy mix integrating clean vehicle deployment and behavioral interventions can mitigate the economic pressure from investment in advanced technologies. Third, there was a spatially differentiated policy effectiveness and feasibility when implementing the low-carbon policy measures, which were due to the different development stages, travel patterns, types of transport systems, vehicle technologies, and energy mixes across China’s 31 provinces. Therefore, a synthetically designed policy package integrating the A-S-I strategies and corresponding to the regional disparities of the implementation effects of low-carbon policy measures is needed.”

Reviewer 2’s specific comments 5)

Minor comments: (1) The authors may want to motivate readers in the introduction part by highlighting the importance of the transport sector in China's national GHG emissions.

Response: Thank you very much for the comments. We have added more text to the Introduction Section to highlight the importance of China’s transport sector in achieving carbon neutrality.

“China’s transport sector contributed 901 million tons to CO₂ emissions in 2019,

accounting for 10% of all CO₂ emissions nationwide (IEA, 2020). From 1990 to 2019, there has been a nearly ten-fold increase in transport-related CO₂ emissions from 94 million tons in 1990 alongside the vigorous development of the Chinese transport industry.”

Reviewer 2’s specific comments 6)

Minor comments: (2) I would suggest "price" instead of economy as one of the four instruments

Response: Thank you very much for the comments. The term “economy” has been changed to “price” in the main text and all figures.

Reviewer 2’s specific comments 7)

Minor comments: (3) energy efficiency should not be confused with energy intensity (of service). See lines 188 and 192 on page 8 and axes in Figure 4.

Response: Thank you very much for your correction. We have modified the corresponding text in the main body as well as in the figure from “energy efficiency” to “energy intensity” to avoid confusion.

Reviewer 2’s specific comments 8)

Minor comments: (4) How do we know "policy effectiveness might be due to economic development at the local level"? (line 230 on page 10) Provide evidence.

Response: Thank you very much for the comments. We admit that this sentence is speculative, so we have revised the text of this sentence to avoid ambiguity.

“Regional disparities in the impacts of each policy intervention on emission reductions indicated that policies on traffic management, information technology, and pricing instrument may be more suitable for implementation in developed regions, while policies for shifting modal structure, promoting new technologies, and releasing regulations offer promising options for developing regions.”

Reviewer 3’s general comments

This paper is intended to depict the long-term pathways to deep decarbonization of China’s transport sector by using a regional transport-energy integrated model. Overall, the methodology and scenario simulations based on the Avoid–Shift–Improve framework are well elaborated and, as a whole, easy to follow, particularly with growing interest in the deep decarbonization of transport sector. I outline some specific concerns below.

Response: We sincerely thank the reviewer for the constructive comments and suggestions, which helped us to substantially improve our manuscript. Please find the point-to-point responses to nine specific comments listed below.

Reviewer 3's specific comments 1)

(1) Regarding the CO₂ emission pathways in Fig.3, it would be better to provide more details on how to set the combinations of 12 scenarios.

Response: Thank you very much for raising this issue. We have expanded the explanation in the Methods section to clarify how the combinations of 12 scenarios were set, as follows:

“In addition to each individual scenario designed based on the A-S-I approach, a series of combined scenarios were created to identify the emission reduction effects of different strategies and instruments. All instruments under the same strategy were combined for the scenarios of “Avoid”, “Shift”, and “Improve”, while the same instruments under different strategies were grouped into the combined scenarios of “Technology”, “Regulation”, “Information”, and “Price”.”

Reviewer 3's specific comments 2)

(2) In Fig. 4, the CO₂ emissions decreased significantly from 2015 to 2060 in the BaU scenario without implementing any Avoid–Shift–Improve policies. I think it should be more clearly indicated how the storyline of BaU scenario was constructed for this study.

Response: Thank you very much for pointing out this problem. Although the low-carbon policy measures designed based on the Avoid–Shift–Improve framework were not implemented in the BaU scenario, CO₂ emissions still decreased significantly from 2015 to 2060 because of the decreasing population in China and the gradual penetration of energy-efficient technologies in the transport sector. Following this comment, we have added an explanation in the section on scenario settings to clearly indicate how the storyline of the BaU scenario was constructed:

“Furthermore, a BaU scenario, which presumed the continuation of technological improvements at the current pace and maintenance of the existing transport and energy policies, was designed to explore the default pathways of energy transition and emission reduction in China's transport sector. In particular, the setting of technological development in the BaU scenario is mainly based on the data and assumptions made in China's Development Plan for the New Energy Automobile Industry. By comparing the scenarios that assume exogenous monetary policy shocks with the BaU scenario, the impacts of potential low-carbon options would be investigated to propose a synthetically designed policy package toward carbon neutrality by the mid-21st century.”

Reviewer 3's specific comments 3)

(3) Since Fig. 5 contains multi-panels, it is recommended to add panel titles like a, b, c for heatmap, GIS map, and scatter diagram.

Response: Thank you very much for the comments. We have added panel titles for the heatmap, GIS map, and scatter diagram in Figure 5.

Reviewer 3's specific comments 4)

(4) When analyzing the regional disparities in policy impacts, it is not clear to an international reader where the provinces are located in China and why some of the provinces can be described as developing or developed. Can the authors provide an additional information and metric for developing and developed regions?

Response: Thank you very much for mentioning this concern. We added Supplementary Figure 3 to clearly show the economic level in terms of GDP per capita in 31 provinces.

Reviewer 3's specific comments 5)

(5) Regarding the economic costs in Fig. 6, I think it needs to be indicated somewhere whether the infrastructure cost has been included.

Response: We thank the reviewer for pointing this out. The economic costs in Figure 6 include the investments for transport devices (e.g., vehicles) but exclude the infrastructure costs. We agree with the reviewer's opinion that infrastructure for low-carbon transport development is important. The decarbonization of the transport sector would be impossible without appropriate infrastructure such as electric quick-charging stations and public transit systems. We have added text in the Discussion and Conclusion section to note this limitation that should be addressed in future work, as follows:

“In addition, the decarbonization of the transport sector would be impossible without appropriate infrastructure such as electric quick-charging stations, but transport infrastructure costs were not included in the model, which needs more attention when improving the modal structure in the future.”

Reviewer 3's specific comments 6)

(6) The pandemic will exert a profound effect on the decarbonization of China's transport sector. Did the authors consider the impacts of the pandemic on GDP, population, mobility, EV production, and behaviors in this study?

Response: Thank you very much for highlighting this issue. It is true that the COVID-19 pandemic will likely influence the transport system significantly. However, given the rapid onset of the pandemic, we have not yet collected reliable long-term data on the transport demand and socioeconomic impacts of the pandemic. We agree there is a need to consider this impact in future work. We have added this point in describing the roadmap for future work, as follows:

“Also, the scenarios were designed without considering the potential effects of the global coronavirus disease pandemic on economic growth and mobility, which warrants future attention if reliable data are available.”

Reviewer 3's specific comments 7)

(7) With respect to the methodology, it would be better to provide a diagram for more details on how the model works, and particularly how the transport model was coupled with an energy system model, at least in the supplementary information.

Response: We thank the reviewer for pointing this out. We have added Supplementary Figure 1 to provide a diagram of the model structure.

Reviewer 3's specific comments 8)

(8) Why did you not consider air and water transportation? They consume a lot of liquid fossil fuels and generate a massive amount of carbon emissions.

Response: Thank you very much for highlighting this issue. This paper is intended to develop a regional transport energy model at the provincial level for China's ground transport sector, excluding air and water transportation. We agree with the reviewer's comment that air and water transportation consume lots of fuel and generate a massive amount of carbon emissions, but it is difficult to collect the air and water transport demand data for air transportation for each province, as the data for domestic air and water transportation is often presented at the national instead of provincial level. According to this comment, the following text has been added to highlight these limitations and potential future work.

“The integrated transport energy model in this paper mainly involved China's ground transport sector that accounts for an overwhelming proportion of the total transport-related CO₂ emissions but did not include air and water transport due to the lack of reliable data at the regional level in China.”

Reviewer 3's specific comments 9)

(9) How about the advanced technologies such as personal aerial vehicles, package delivery drones, robots for delivery? Will the transport sector be influenced by the technological innovations in the future?

Response: Thank you very much for pointing out this problem. Regarding advances in technology, new technologies such as hybrid EVs, plug-in hybrid EVs, battery EVs, and FCVs have been considered in the transport energy model, but we did not incorporate technologies that have not been popularized and are not likely to be applied reliably in the transport sector. We have added a discussion of limitations and future work, as follows:

“Moreover, future research is required to further clarify the role of the transport sector in achieving carbon neutrality. This study only considered transport technologies that are currently in operation or are likely to be popularized in the near future but did not account for some new technologies such as personal aerial vehicles, self-driven cars and trucks, package delivery drones, and delivery robots, which will not be applied reliably without

a major technological breakthrough.”

Reviewer comments, second round review

Reviewer #1 (Remarks to the Author):

I thank the authors for their comprehensive responses and revisions. I still find that the paper structure can be improved such that the methodology is presented before the conclusions, but if the journal requires that the format be this way, then that's fine.

Reviewer #2 (Remarks to the Author):

The authors successfully addressed nearly all of the earlier comments I made in my referee report. In particular, it is impressive to see how they addressed my critical points on potential synergistic coupling (or trade-offs) between strategies and instruments for carbon neutrality. So, I think this manuscript is in a better shape now, except two minor suggestions. (1) The second paragraph in Discussion and Conclusion section can be improved. In particular, the sentence starting with "Meanwhile," is not easily understood as economic costs already encompasses financial costs. (2) The difference between Figure 2 and Figure 3 is not very much clear without giving a second thought. The authors claim Figure 3 "combines" the 12 scenarios in Figure 2 into three strategies and four instruments. Can you articulate what you mean by "combines" in the text?

Reviewer #3 (Remarks to the Author):

I am OK with the revision. Thanks and regards.

Point-by-point responses:

Reviewer 2's comments 1)

The authors successfully addressed nearly all of the earlier comments I made in my referee report. In particular, it is impressive to see how they addressed my critical points on potential synergistic coupling (or trade-offs) between strategies and instruments for carbon neutrality. So, I think this manuscript is in a better shape now, except two minor suggestions. (1) The second paragraph in Discussion and Conclusion section can be improved. In particular, the sentence starting with "Meanwhile," is not easily understood as economic costs already encompasses financial costs.

Response: Thank you very much for pointing up this issue. What we were mainly trying to say was that the economic costs include the investment costs of new technologies and devices, as well as the costs of behavioral change. To avoid the confusion between the two terms “economic costs” and “financial costs”, we uniformly revised them to “device costs” and “behavioral costs” and revised the text of this sentence as follows:

“Meanwhile, the positive impacts on economic costs of technology improvement related scenarios would be overestimated because the device costs of investing in new low-carbon technologies would decrease with the reduction of transport demand and positive modal shift, but the behavioral costs that contributed to transport demand reduction and modal shift were not included.”

Reviewer 2's comments 2)

(2) The difference between Figure 2 and Figure 3 is not very much clear without giving a second thought. The authors claim Figure 3 "combines" the 12 scenarios in Figure 2 into three strategies and four instruments. Can you articulate what you mean by "combines" in the text?

Response: Thank you very much for the comments. We have added more text in this section to clarify how we combined 12 scenarios in Figure 2 into three strategies and four instruments in Figure 3, as follows:

“All instrument scenarios under the same strategy category were combined as “Avoid”, “Shift”, and “Improve” scenarios, while the same instruments under different strategies were combined as “Technology”, “Regulation”, “Information”, and “Price” scenarios.”

Moreover, we have also added more explanations in the Methods section to articulate how to set the combination scenarios, as follows:

“In addition to each individual scenario designed based on the A-S-I approach, a series of combined scenarios were created to identify the emission reduction effects of different strategies and instruments. All instruments under the same strategy were combined for

the scenarios of “Avoid”, “Shift”, and “Improve”, while the same instruments under different strategies were grouped into the combined scenarios of “Technology”, “Regulation”, “Information”, and “Price”. In these combined scenarios, all assumptions and parameters under the same strategy or instrument category are given simultaneously.”